# Modulation of ATP Production Influences Inorganic Polyphosphate Levels in Non-Athletes’ Platelets at the Resting State

**DOI:** 10.3390/ijms231911293

**Published:** 2022-09-25

**Authors:** Takashi Ushiki, Tomoharu Mochizuki, Katsuya Suzuki, Masami Kamimura, Hajime Ishiguro, Tatsuya Suwabe, Tomoyuki Kawase

**Affiliations:** 1Division of Hematology and Oncology, Graduate School of Health Sciences, Niigata University, Niigata 951-8518, Japan; 2Department of Transfusion Medicine, Cell Therapy and Regenerative Medicine, Niigata University Medical and Dental Hospital, Niigata 951-8520, Japan; 3Department of Hematology, Endocrinology and Metabolism, Faculty of Medicine, Niigata University, Niigata 951-8510, Japan; 4Department of Orthopaedic Surgery, Graduate School of Medical and Dental Sciences, Niigata University, Niigata 951-8510, Japan; 5Division of Oral Bioengineering, Graduate School of Medical and Dental Sciences, Niigata University, Niigata 951-8514, Japan

**Keywords:** platelet, polyphosphate, ATP, NADH, oxidative phosphorylation

## Abstract

Platelets produce inorganic polyphosphate (polyP) upon activation to stimulate blood coagulation. Some researchers have linked polyP metabolism to ATP production, although the metabolic linkage is yet to be elucidated. We found evidence for this possibility in our previous study on professional athletes (versus non-athletes), and proposed that the regulatory mechanism might be different for these two groups. To explore this aspect further, we investigated the effects of modulated ATP production on polyP levels. Blood samples were obtained from Japanese healthy, non-athletes in the presence of acid-citrate-dextrose. The platelets in the plasma were treated with oligomycin, rotenone, and GlutaMAX to modulate ATP production. PolyP level was quantified fluorometrically and visualized using 4′,6-diamidino-2-phenylindole. Correlations between polyP and ATP or NADH were then calculated. Contrary to the hypothesis, inhibitors of ATP production increased polyP levels, whereas amino acid supplementation produced the opposite effect. In general, however, polyP levels were positively correlated with ATP levels and negatively correlated with NADH levels. Since platelets are metabolically active, they exhibit high levels of ATP turnover rate. Therefore, these findings suggest that ATP may be involved in polyP production in the resting platelets of non-athletes.

## 1. Introduction

Inorganic polyphosphate (polyP) is a polymer of linear chains of orthophosphate residues linked by high-energy phospho–anhydride bonds [1]. PolyP has been detected in several subcellular compartments of mammalian cells, including the cytoplasm, plasma membranes, and organelles such as lysosomes, dense granules, mitochondria, and nuclei [2,3]. The major functions of polyP in mammalian cells include cell proliferation, angiogenesis, apoptosis, osteoblast function, bone mineralization, energy metabolism, tumor metastasis, blood clotting, inflammation, cellular bioenergetics, signal transduction, Ca^2+^ signaling, neuronal excitability, and wound healing [1,3].

It is generally believed that polyP is constantly produced and maintained at sufficient levels to respond rapidly to external stimuli and regulate multiple functions promptly. Platelets have significantly higher levels of ATP turnover than other cell types, even when they are quiescent, because of their higher metabolic activity [4,5]. Mitochondria are critical for aerobic respiration and provide the metabolic substrates required for platelet function and survival. Consequently, mitochondria determine platelet longevity (7–10 days) [5]. However, it is believed that the few (5–8) mitochondria found in platelets are small and have few cristae [6]. These mitochondria are not fully capable of meeting the energy output needed to support the platelets’ multiple functions in the activation state. This necessitates the existence of multiple energy-transducing pathways [4]. A recent study proposed the existence of extra-mitochondrial machinery for energy production [6] as a remedy for possible energy shortages.

Existing analytical tools for quantifying polyP levels and visualizing polyP localization have several shortcomings, including low specificity. However, recent technological advances using a fluorescent dye, 4′,6-diamidino-2-phenylindole (DAPI), have made high-throughput assays possible. In previous studies [7,8], the DAPI-dependent assay technique was improved, and its usefulness and reliability were validated. In a subsequent study, we found [9] that platelet polyP might suppress the SARS-CoV-2-mRNA vaccine-induced adverse effects following the first vaccination in young female participants, but not following the second vaccination. More recently, we found that platelets in professional athletes who exhibit higher metabolic activity than non-athletes may display a distinct regulatory system for maintaining the ATP–polyP balance, because polyP levels are correlated with cytoplasmic NADH levels [10]. This finding has motivated further studies to better understand the mechanism underlying the potential conversion of these energy-rich molecules.

A strong relationship between cellular ATP levels and polyP levels has been observed in numerous types of mammalian cells, including fibroblasts, kidney cells, and adrenal cells [2]. Here, we examined the effects of modulators of ATP production on polyP levels in the platelets of non-athletes at rest to expand our previous findings and integrate them with the current understanding. In addition, we visualized the cellular localization of polyP to validate the biochemical results.

## 2. Results

### 2.1. Biochemical and Fluorometric Quantification of Platelet PolyP

Oligomycin is a well-known inhibitor of mitochondrial F_0_F_1_-ATP synthase [11,12]; hence; it is expected to reduce platelet ATP levels. The effects of oligomycin on platelet polyP, ATP, and NADH levels are shown in Figure 1. As expected, platelet ATP levels decreased dose-dependently after 6 h but not after 20 h of treatment. Conversely, NADH levels increased in the oligomycin-treated platelets after 6 and 20 h of treatment. PolyP levels did not significantly alter after 6 h, but these levels gradually, but not significantly, increased in a dose-dependent manner after 20 h.

Based on previous studies [13,14], to confirm the inhibitory effects of oligomycin on mitochondrial respiration and consequent ATP production, the mitochondrial membrane potential (ΔΨM) was examined semi-quantitatively. The effects of oligomycin (100 μM) on fluorescence intensity at 6 and 20 h are shown in Figure 2. Oligomycin substantially reduced the ΔΨM; this inhibitory effect lasted for 20 h.

Since rotenone impairs the mitochondrial electron transport chain [12,13], this inhibitor likely reduces platelet ATP levels. The effects of rotenone on platelet polyP, ATP, and NADH levels are shown in Figure 3. Neither the platelet ATP nor NADH levels changed significantly. Furthermore, PolyP levels increased significantly only in response to the lower dose after 6 h; no significant effects were observed at higher doses or after 20 h of treatment.

To confirm the inhibitory effects of rotenone on mitochondrial respiration and the consequent ATP production, ΔΨM was examined again, similarly to the case with oligomycin. The effects of rotenone (200 μM) on fluorescence intensity at 6 and 20 h are shown in Figure 4. As observed with oligomycin, rotenone substantially reduced ΔΨM; this inhibitory effect lasted for 20 h.

The images associated with Figure 2 and Figure 4 were image analyzed and the brightness was quantified and normalized by particle as shown in Table 1. Both oligomycin (100 μM) and Rotenone (200 μM) reduced these values: however, a significant difference was observed only in oligomycin. 

The images associated with Figure 2 and Figure 4 were analyzed and quantified. Data are expressed as the mean ± standard deviation. Since the sample size was small (n = 4), non-parametric analyses were performed to compare the mean values via the Kruskal–Wallis one-way analysis of variance, followed by a Tukey multiple comparisons test.

GlutaMAX is an l-alanyl-l-glutamine dipeptide that does not degrade into ammonia during prolonged storage or incubation [14]. GlutaMAX supplementation can be used as a direct substitute for L-glutamine, which drives oxidative phosphorylation (OXPHOS) to produce ATP [15]. The effects of GlutaMAX on platelet polyP, ATP, and NADH levels are shown in Figure 5. Neither the platelet ATP nor NADH levels changed significantly. PolyP levels significantly decreased after 20 h but not after 6 h of treatment.

The correlations between polyP and ATP or NADH levels and between ATP and NADH levels in platelets treated with oligomycin, rotenone, or GlutaMAX for 20 h are shown in Figure 6. There were weak-to-strong positive correlations between ATP and polyP, regardless of the reagent added. These correlations were statistically significant in platelets treated with rotenone or GlutaMAX. In contrast, weak-to-strong negative correlations were observed between NADH and polyP. This correlation was statistically significant only in oligomycin-treated platelets. However, regardless of the reagent added, the correlations between ATP and NADH were either weakly negative or absent.

### 2.2. Cytochemical Visualization of Platelet PolyP

The DAPI-positive molecules found in platelets could be categorized into two types: intensely stained microparticles, and faintly stained cytoplasm or the plasma membrane (Figure 7). Fluorometric analysis can be used to detect and integrate the fluorescence intensity of reactive molecules to quantify platelet polyP concentration. In the first half of this section, we used decalcification reagents and alkaline phosphatase (ALP) to examine which of these was more likely to have a high polyP concentration.

The effects of Kalkitox and EDTA on platelet polyP are shown in Figure 8. Kalkitox and EDTA were used to decalcify tissue samples to prepare paraffin-embedded specimens. Short-term treatment with undiluted Kalkitox or diluted EDTA (5 mM) significantly reduced the number of DAPI-reactive microparticles, whereas the reduction was less pronounced in faint staining. To easily recognize and compare the DAPI-reactive microparticles among the images, the right column of Figure 5 shows the black and white images converted from the red channel of the RGB images.

The effects of ALP on platelet polyP levels are shown in Figure 6. In the previous study [7], the effects of ALP were examined and reported. However, because the effectiveness of ALP was limited, we re-examined this issue by altering the source of ALP. The specific alkaline buffer connected to the newly obtained ALP significantly reduced the number of DAPI-reactive microparticles (without ALP) after 24-h treatment. Therefore, the treatment duration was decreased to 8 h, and the net ALP effects were evaluated. Similar to what has been described previously [7], ALP reduced the number of DAPI-reactive microparticles. Based on their high fluorescence intensity, that is, the bulk of the fluorescence signal, and sensitivity to decalcification and ALP, these microparticles are probably the predominant polyP in resting platelets. Therefore, in subsequent experiments, we focused on these microparticles.

In the second half of this section, the biochemical data were validated using cytochemical analysis. The effects of oligomycin on platelet polyP levels are shown in Figure 9. As previously observed (Figure 1), 20-h treatment with oligomycin substantially increased the number of DAPI-reactive microparticles.

The effects of rotenone on platelet polyP levels are shown in Figure 10. In support of the quantified data shown in Figure 2, 20-h treatment with rotenone (20 μM) significantly increased the number of DAPI-reactive microparticles.

The effects of GlutaMAX on platelet polyP levels are shown in Figure 11. Again, the quantified data shown in Figure 3 is supported by the observation that the 20-h GlutaMAX (10%) treatment substantially decreased the number of DAPI-reactive microparticles.

## 3. Discussion

Platelet polyP cellular metabolism, subcellular localization, and individual variability have all been poorly investigated because of the lack of clarity and controversy around the analytical methodologies. In a recent study [10] using modified DAPI-based methods, we suggested that energy metabolism, that is, the reciprocal conversion between ATP and polyP, could vary between professional athletes and non-athletes. We studied the impact of metabolic inhibitors and energy supplements on polyP levels to further explore the mechanism underlying this difference.

### 3.1. DAPI-Stained Microparticles

Similar to what we reported earlier, the present biochemical and cytochemical analysis confirmed that microparticles constitute the primary forms of platelet polyP [7]. Most polyP in platelets is encapsulated in amorphous nano/microparticles and associated with the exposed membrane surfaces. Anionic polyP, unlike orthophosphate and calcium, readily binds to mono-atomic cations, like Ca^2+^, to produce Ca-polyP microparticles via reversible bonds [11,16]. In contrast, mammalian ALP is a potent exopolyphosphatase found on the cell surface, which catalyzes polyP to AMP by sequentially removing terminal phosphates from polyP [3]. Thus, our findings on decalcification agents and ALP suggest that DAPI-reactive microparticles are the primary form of platelet polyP.

However, the precise localization (intracellular or extracellular space, including plasma membrane surface) has not yet been identified. Complementary cytochemical analyses revealed identical responses to exogenous reagents—reduction in ATP production increased polyP levels, whereas the acceleration of ATP production decreased polyP levels in the resting platelets of non-athletes. Due to minimal non-specific binding to cell-surface proteoglycans and other biopolymers, the quantities observed may be slightly higher than the net polyP levels. However, we believe these data are sufficiently reliable for a reasonable interpretation and further discussion.

### 3.2. Energy Metabolism and PolyP Turnover at Cellular Levels

In contrast to our findings, Pavlov et al. reported in 2010 that polyP levels were increased by respiration substrates and reduced by rotenone, and that oligomycin inhibited polyP formation [17]. Thus, they hypothesized that polyP directly contributes to ATP production or serves as a reservoir for high-energy phosphoryl groups to regulate enzymatic activity rates. Several research groups subsequently supported this finding [18,19]. 

At the cellular level, we speculate that the inconsistent or contradictory results could be due to homeostatic control or metabolic flexibility, such as the switch from oxygen-dependent to glycolytic ATP synthesis under hypoxic conditions. Consequently, the likely explanation for GlutaMAX not significantly enhancing platelet ATP levels could be as follows. Glucose, glutamine, and fatty acids are primary energy substrates that contribute to ATP production [4]. Among these substrates, the role of glucose in ATP turnover is 3.6%, similar to that of glutamine and much lower than that of palmitate and other carbon sources [4]. Considering this limited share, even if GlutaMAX significantly increased glutamine availability for cellular ATP production, this supplementation may not significantly increase total ATP levels. 

Furthermore, the chemical and biological stability of polyP and ATP should also be considered. ATP is chemically unstable and is degraded rapidly by various phosphatases. PolyP (particularly medium or short length) is thought to be chemically more stable than ATP. However, the cellular levels of polyP are highly dynamic and respond rapidly to various inhibitors and activators of mitochondrial respiration [6]. PolyP production is closely related to the energetic state of the mitochondria, that is, the presence of ATP and the activity of mitochondrial F_0_F_1_-ATP synthase [2,17,18,19,20]. ATP synthase is a key enzyme in polyP hydrolysis. The abundance of polyP interacts with ADP during mitochondrial respiration and OXPHOS and substantially increases ATP production in mitochondria [19]. Therefore, it can be inferred that the polyP level regulates the cellular ATP level and vice versa [17,19]. In addition, due to cell-surface exopolyphosphatase, polyP has a half-life of approximately 1.5–2 h in human blood or plasma [3]. Although it is challenging to quantify polyP by appropriate normalization, cytochemical visualization may be a more effective method for capturing polyP since it employs rapid inactivation of the respective enzymes.

### 3.3. Possible Involvement of ATP in PolyP Production

Reciprocal conversion between ATP and polyP is thought to occur depending on the metabolic activity. This possibility was observed in a previous study of professional athletes [10]. We also found that such reciprocal conversion rarely occurs in the platelets of non-athletes at rest. Thus, we hypothesized that this concept could be applied to any human subject, regardless of athletic training. It may vary based on the initial platelet specification or activation status of individuals. Thus, when high ATP levels are required to maintain cellular functions, polyP serves as an energy source to produce ATP without intermediate steps or oxygen [18]. 

Based on our current understanding of ATP, NADH, and polyP metabolism, we present a schematic diagram of our proposed mechanism for ATP-to-polyP conversion in Figure 12. Briefly, in platelets, ATP is generated mainly from glucose and also from glutamine through glycolysis and OXPHOS. When ATP is actively generated, the platelet NADH level decreases. A portion of the generated ATP is immediately used for energy-dependent cellular processes, while the rest is stored as ATP and polyP in the dense granules. When ATP is depleted, this polyP is cleaved to ATP to function as conventional energy fuel, and is also presumably capable of directly providing energy. The OXPHOS machinery is mainly localized in the mitochondria. However, it has also been suggested that the OXPHOS machinery is expressed in platelet membranous structures other than mitochondria [6]. This ectopic presence is needed to meet the higher energy demand of platelets, which are involved in both primary hemostasis and antimicrobial host defense [21].

### 3.4. Clinical Relevance

To better understand the unbalanced bioenergetics of dysfunctional mitochondria, it is important to elucidate the regulatory mechanism of polyP turnover and its pathophysiological involvement in mammalian bioenergetics. In addition, considering the stimulatory effects of polyP on blood coagulation, this approach is expected to improve platelet-rich plasma therapy, which is frequently used in regenerative medicine [22,23].

## 4. Materials and Methods

### 4.1. Reagents 

Oligomycin (LKT Labs, St. Paul, MN, USA) was dissolved in dimethyl sulfoxide (DMSO) to produce a stock solution with a final concentration of 10 mM and stored at −20 °C. Immediately before use, the stock solution was diluted with dimethyl sulfoxide (DMSO) and added to the platelet suspension. In the control, the same volume of DMSO (without oligomycin) was added as described above. Rotenone (Merck, Darmstadt, Germany) was dissolved in dimethyl sulfoxide (DMSO) to yield a 200 mM stock solution. This solution was frozen, diluted, and applied to a platelet suspension similar to oligomycin. Appropriate amounts of GlutaMAX (Thermo Fisher Scientific, Waltham, MA, USA) were directly added to the platelet suspension without dilution. The Cellstain^®^ DAPI solution, ATP assay kit-luminescence, and Cell Counting kit-8 (WST-8) were obtained from Dojindo Molecular Technologies, Inc. (Kumamoto, Japan) and used according to the manufacturer’s instructions. Calf intestine-derived alkaline phosphatase (ALP) was obtained from TOYOBO (Osaka, Japan). Kalkitox and EDTA (0.5 mol/L, pH = 7.4) were purchased from FUJIFILM Wako Pure Chemical Corp. (Osaka, Japan).

### 4.2. Blood Cell Counting

Blood cell counts were performed using an automated hematology analyzer (pocHi V-diff, Sysmex Corporation, Kobe, Japan) before centrifugation and measuring platelet polyP levels to prepare the platelet suspension and adjust platelet counts, respectively [8,9]. If the histograms of platelet distribution did not display a smooth curve, the samples were discarded and not subjected to further polyP quantification.

### 4.3. Preparation of Pure Platelet-Rich Plasma and Treatments

The study design and consent forms for all procedures (project identification code: 2021-0126) were approved by the Ethics Committee for Human Participants at Niigata University (Niigata, Japan) and complied with the Helsinki Declaration of 1964, as revised in 2013.

Blood samples were collected from six non-smoking, healthy male (n = 15; ages 27–62 year) and female volunteers (n = 3; ages 43–66 year. Despite having lifestyle-related diseases and taking medications, these donors (i.e., our team members and relatives) had no limitations in their daily activities. These donors were also negative for HIV, HBV, HCV, or syphilis infections. A prothrombin test was performed on all blood samples using a CoaguChek^®^ XS (Roche, Basel, Switzerland), and all results were normal. Platelet and other blood cell counts were measured using a pocH 100iV automated hematology analyzer (Sysmex, Kobe, Japan). Approximately 9 mL of peripheral blood was collected in plain glass vacuum blood collection tubes (Vacutainer^®^; BD Biosciences, Franklin Lakes, NJ, USA) containing 1.5 mL of acid-citrate-dextrose solution. Whole blood samples were stored in a rotating agitator at an ambient temperature (22–25 °C) for 24 h. The samples were centrifuged horizontally at 415× *g* for 10 min (soft spin) (Kubota, Tokyo, Japan). The upper plasma fraction, which was ~2 mm beyond the interface of the plasma and red blood cell fractions, was transferred into 2 mL sample tubes and incubated with the inhibitors or supplement for 6 or 20 h at room temperature. At the end of incubation, the samples were centrifuged at 664× *g* for 4 min (hard spin) using an angle-type centrifuge (Sigma Laborzentrifugen, Osterode am Harz, Germany) to collect the resting platelet pellets. Then, platelets were resuspended in phosphate-buffered saline (PBS) to adjust the platelet concentration to 3.0 × 10^5^/µL.

In accordance with guidelines from the ethics committee, two blood collection tubes were used for each donor, and each sample was randomly allocated to one of the treatment groups.

### 4.4. Determination of PolyP Levels

Briefly, the fixed platelets were centrifuged, and the resulting platelet pellets were gently suspended in Milli-Q water. Then, an automated hematology analyzer was used to determine the blood cell counts. Samples within the optimized range were incubated with 4 μL/sample (4 μg/mL) DAPI for 30 min at room temperature (22–25 °C) and directly subjected to fluorescence measurements using a fluorometer (FC-1; Tokai Optical Co., Ltd., Okazaki, Japan) at excitation and emission wavelengths of 425 and 525 nm, respectively [8,9].

### 4.5. Determination of NADH and ATP Levels

The number of non-fixed, living platelets suspended in PBS (100 μL) was adjusted to a density ranging between 2.5 and 4.0 × 10^7^/100 µL, combined with 10 µL WST-8 solution, and incubated at room temperature (22–25 °C) for 1 h. After centrifugation, the supernatants were analyzed using a spectrophotometer at 450 nm (SmartSpec Plus; Bio-Rad Laboratories, Inc., Hercules, CA, USA), as previously described [8,9].

Platelet ATP levels were determined using a luminescence ATP assay kit and luminescencer (AB-2200, Atto Corp., Tokyo, Japan).

### 4.6. Cytochemical Visualization of Platelet PolyP

PolyP stored in and probably released from platelets was visualized using DAPI. Platelets were immobilized on glass slides for 4 h using a Cytospin 4 cytocentrifuge (Thermo Fisher Scientific, Waltham, MA, USA) and fixed with 10% neutral-buffered formalin. After washing with 0.1% Tween-20-containing PBS and subsequently with Milli-Q water, platelets were stained for 30 min with Milli-Q water containing 4 μg/mL DAPI. Next, the specimens were re-washed with 0.1% Tween-20-containing PBS, followed by Milli-Q water, then mounted using an antifade mounting medium (Vectashield; Vector Laboratories, Burlingame, CA, USA). DAPI and phalloidin were detected using a fluorescence microscope (Eclipse 80i; Nikon) equipped with a BV-2A filter cube (excitation filter: 400–440 nm; dichroic mirror: 455 nm; barrier filter: 470 nm).

### 4.7. Treatments with Decalcification Agents or ALP

The fixed platelets were treated with 0.3 U/μL (1:10) ALP in an ALP-specific buffer at room temperature (22–25 °C) for 8 h. Alternatively, the fixed platelets were treated for 30 min at room temperature with either Kalkitox, a decalcification agent, or 5 mM EDTA. As described above, the specimens were washed with Milli-Q water and stained with DAPI.

### 4.8. Image Analysis for DAPI-Stained Particles

The RGB images were split into separate color channels to increase the contrast between densely stained microparticles and faintly stained cytoplasm or plasma membranes. Since the red channel exhibited the most prominent contrast, it was converted into a black-and-white image (Appendix A). For reference, the emission filter (≥470 nm) was replaced with a red filter (640.0 ± 32.5 nm), and the same specimen was photographed, converted to black and white, and compared to the above image. In this study, images from the red channel of the RGB division were adopted for comparison and presented along with the original RGB images.

### 4.9. Visualization of Mitochondrial Membrane Potential

Platelets treated with oligomycin or rotenone were washed, immobilized on glass slides using Cytospin 4, and fixed with 10% neutral-buffered formalin, as described above. After washing with PBS, according to the manufacturer’s instructions, platelets were stained with the MT-1 MitoMP Detection Kit (Dojindo Molecular Technologies, Inc.) and examined using a fluorescence microscope equipped with a G-2A filter cube (excitation filter, 510–560 nm; dichroic mirror, 570 nm; barrier filter, 590 nm).

### 4.10. Image Analysis for Mitochondrial Membrane Potential

Color images were converted to gray scale images and subjected to image analysis using the WinROOF software (version 2021; Mitani Co., Fukui, Japan). In brief, images were binarized by the automatic conversion function for most images. In the dark images, however, thresholds were set at inflection points. The brightness of each image and the number of particles (platelets) were determined in the same area of the region of interest. The brightness values were normalized by the number of particles.

### 4.11. Statistical Analysis

Box plots were used to compare each value across multiple groups. Kruskal–Wallis one-way ANOVA on ranks was performed to confirm statistical differences in the median and spread (SigmaPlot version 14.5; Systat Software, Inc., San Jose, CA, USA). Spearman’s correlation analysis was used to compare the correlations between the two indices, and SigmaPlot software was used to calculate correlation coefficients. Differences were considered statistically significant at *p* < 0.05. The strength of correlation was rated as follows: 0.8–1.0 (very strong), 0.6–0.79 (strong), 0.4–0.59 (moderate), 0.2–0.39 (weak), 0–0.19 (very weak). For comparison of image-analysis data, a non-parametric analysis was performed to compare the mean values via the Kruskal–Wallis one-way analysis of variance, followed by a Tukey multiple comparisons test. Differences were considered statistically significant at *p* < 0.05.

## 5. Conclusions

We found that platelet polyP levels were positively correlated with ATP levels and negatively correlated with NADH levels. Platelets are metabolically active and exhibit a high ATP turnover rate to maintain their activation state. Thus, our findings suggest that ATP may be involved in polyP production in the platelets of non-athletes at rest.

## Figures and Tables

**Figure 1 ijms-23-11293-f001:**
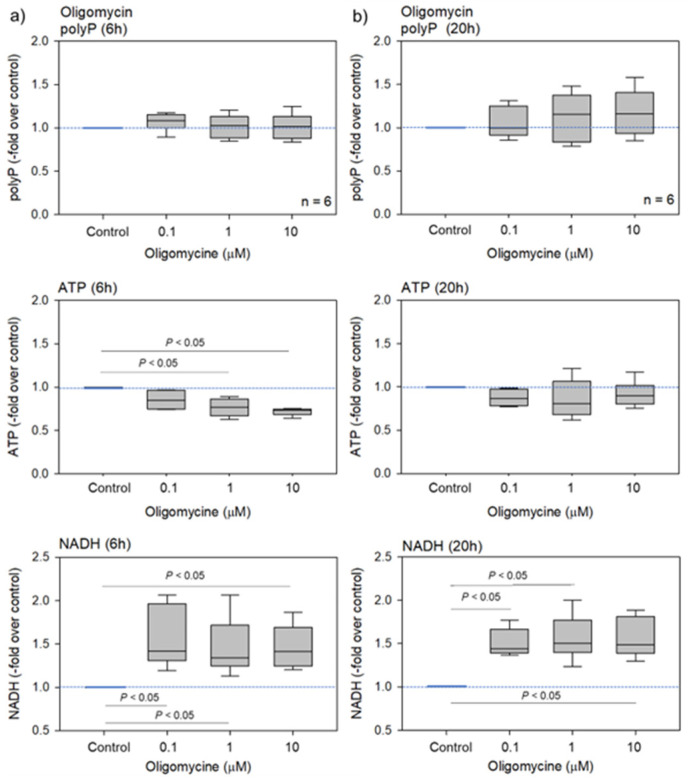
The effects of oligomycin on platelet polyP, ATP, and NADH levels after 6 h (**a**) or 20 h (**b**) of treatment. Statistical analysis was performed using a non-parametric test. n = 6.

**Figure 2 ijms-23-11293-f002:**
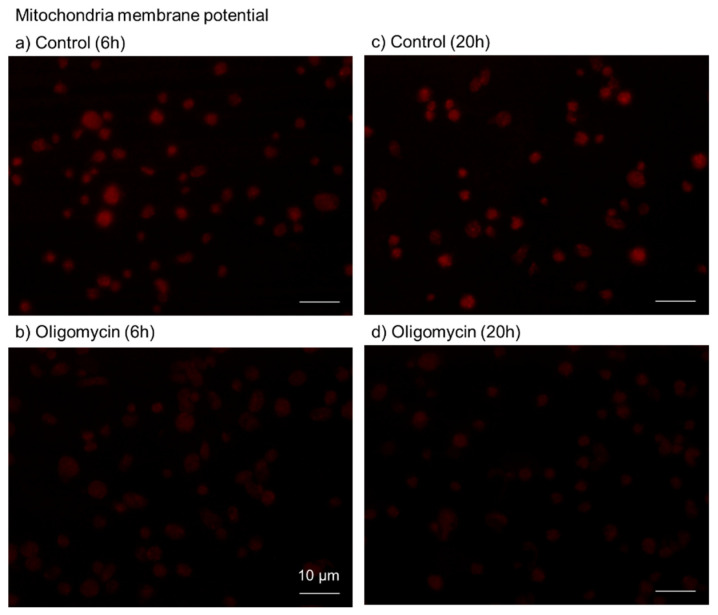
Effects of oligomycin on platelet membrane potential. Platelets were treated with 100 μM oligomycin (**b**,**d**) for 6 h (**a**,**b**) or 20 h (**c**,**d**). The control contained only the solvent (DMSO) (**a**,**c**). Scale bar = 10 μm. Similar results were observed in four other independent experiments.

**Figure 3 ijms-23-11293-f003:**
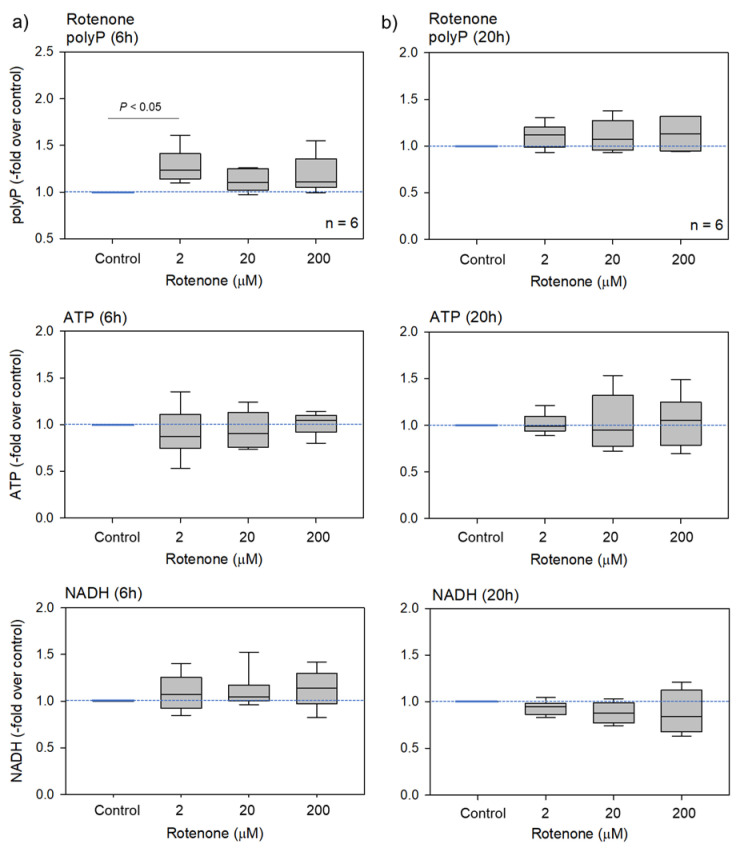
The effects of rotenone on platelet polyP, ATP, and NADH levels after 6 h (**a**) or 20 h (**b**) of treatment. Statistical analysis was performed using a non-parametric test. n = 6.

**Figure 4 ijms-23-11293-f004:**
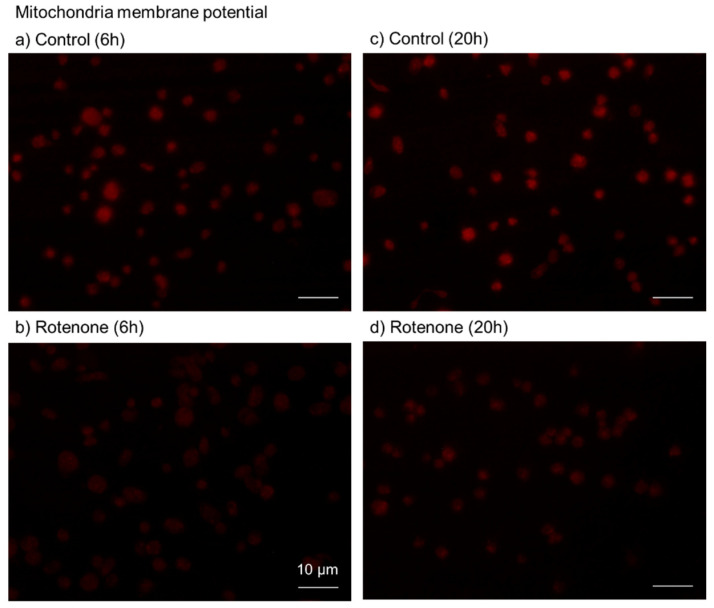
Effects of rotenone on platelet membrane potential. Platelets were treated with 200 μM rotenone (**b**,**d**) for 6 h (**a**,**b**) or 20 h (**c**,**d**). The control contained only the solvent (DMSO) (**a**,**c**). Bar = 10 μm. Similar results were observed in four other independent experiments.

**Figure 5 ijms-23-11293-f005:**
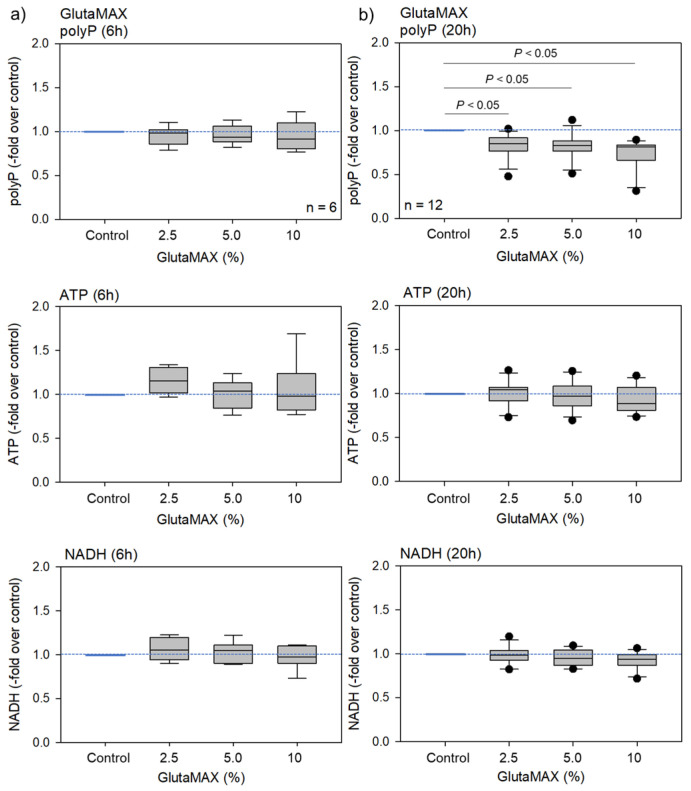
The effects of GlutaMAX on platelet polyP, ATP, and NADH levels after 6 h (**a**) or 20 h (**b**) of treatment. Statistical analysis was performed using a non-parametric test. n = 6 (**a**) or 12 (**b**).

**Figure 6 ijms-23-11293-f006:**
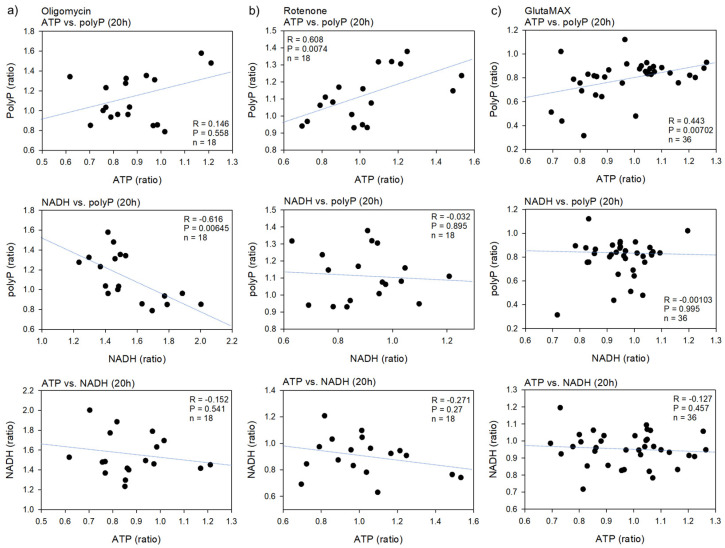
The correlations between polyP levels and ATP levels or NADH levels and between ATP levels and NADH levels in platelets treated with oligomycin (**a**), rotenone (**b**), or GlutaMAX (**c**) for 20 h. Spearman’s rank correlation coefficients were calculated for the evaluation of their correlations.

**Figure 7 ijms-23-11293-f007:**
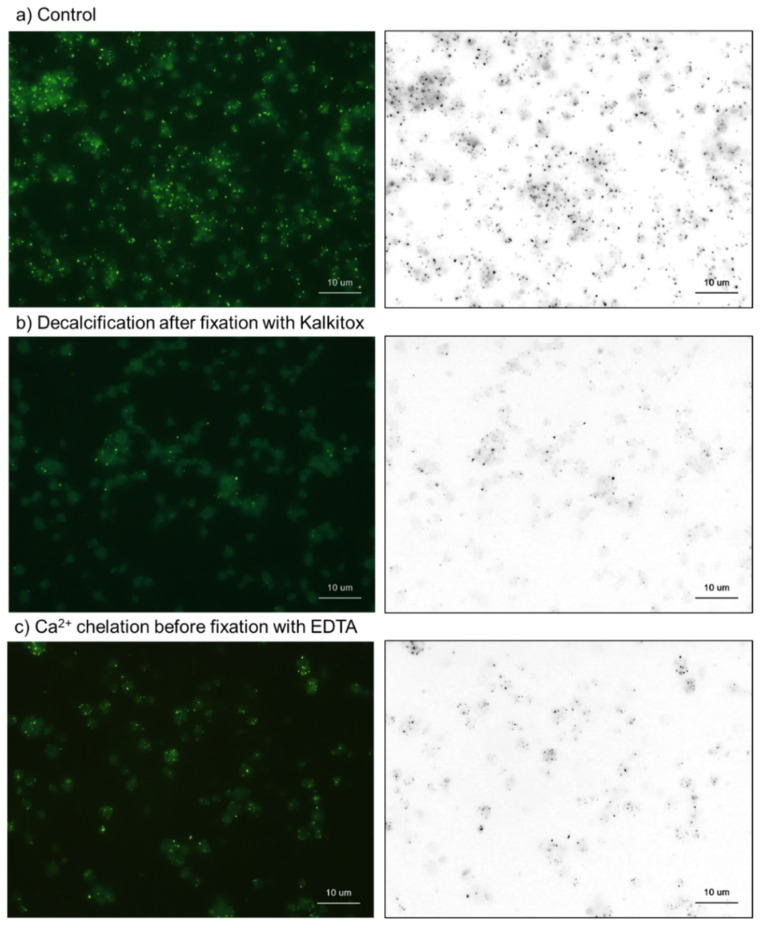
The effects of Kalkitox and EDTA on platelet polyP (green and black microparticles in left and right columns, respectively). Fixed platelets were treated with Kalkitox or 5 mM EDTA for 30 min (**b**). The control was incubated in Milli-Q water without decalcification reagents (**a**). Bar = 10 μm. Similar results were observed in four other independent experiments.

**Figure 8 ijms-23-11293-f008:**
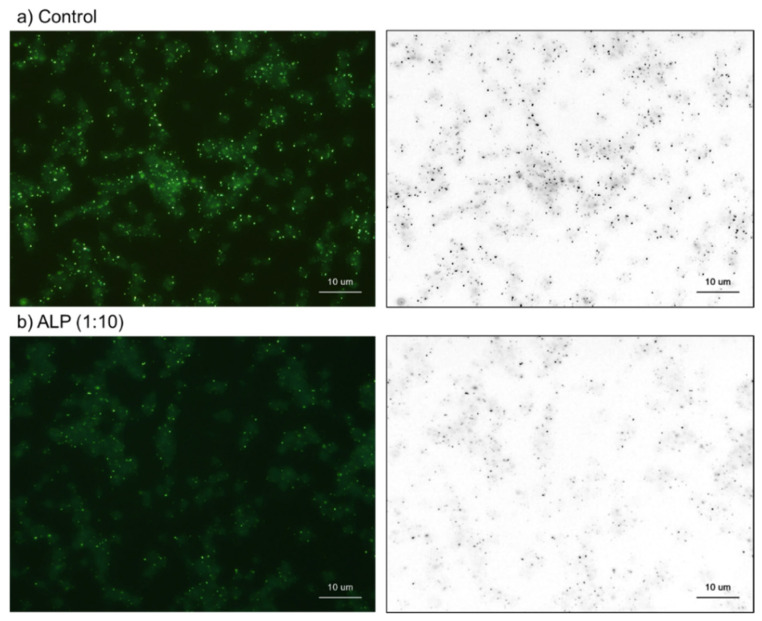
The effects of ALP on platelet polyP. Fixed platelets were treated with 0.3 U/mL (1:10) ALP for 8 h (**b**). The control was incubated in the specific buffer without ALP (**a**). Bar = 10 μm. Similar results were observed in four other independent experiments.

**Figure 9 ijms-23-11293-f009:**
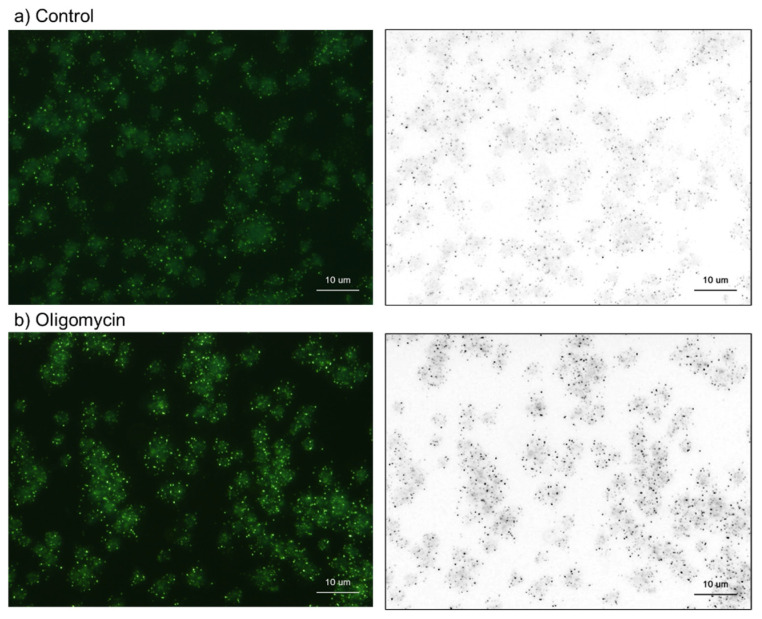
The effects of oligomycin on platelet polyP. Platelets were treated with 10 μM oligomycin (**b**) for 20 h. The control contained only the solvent (DMSO) (**a**). Bar = 10 μm. Similar results were observed in four other independent experiments.

**Figure 10 ijms-23-11293-f010:**
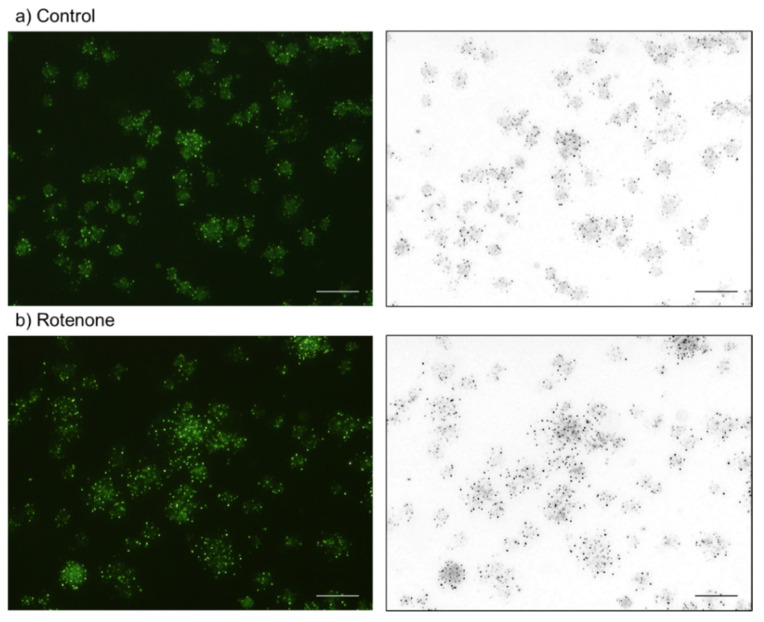
The effects of rotenone on platelet polyP (green and black microparticles in left and right columns, respectively). Platelets were treated with 20 μM rotenone (**b**) for 20 h. The control contained only the solvent (DMSO) (**a**). Bar = 10 μm. Similar results were observed in four other independent experiments.

**Figure 11 ijms-23-11293-f011:**
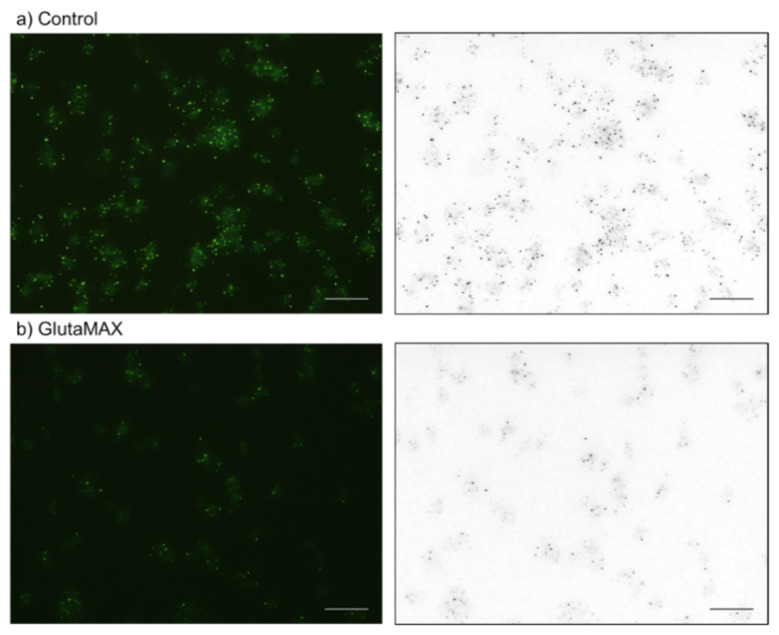
The effects of GlutaMAX on platelet polyP (green and black microparticles in left and right columns, respectively). Platelets were treated with 10% GlutaMAX (**b**) for 20 h. The control had no additions (**a**). Bar = 10 μm. Similar results were observed in four other independent experiments.

**Figure 12 ijms-23-11293-f012:**
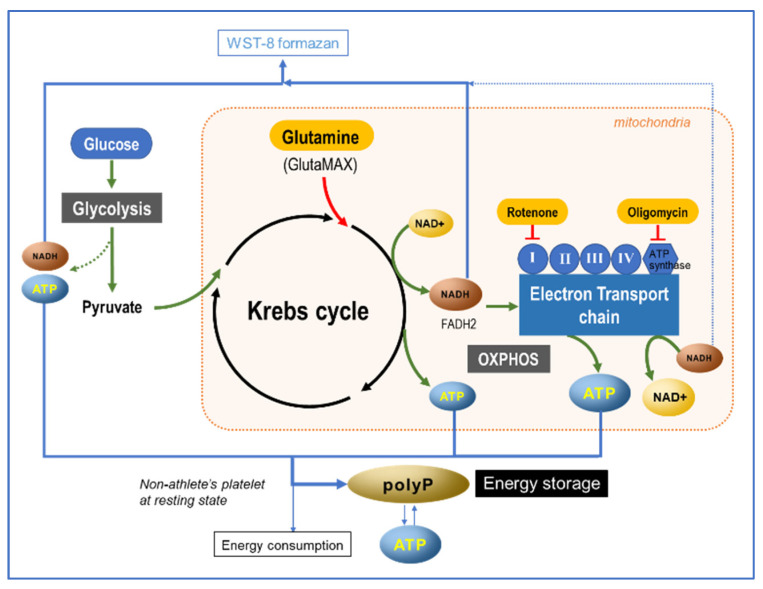
Proposed mechanism of polyP conversion from ATP in resting platelets of non-athletes.

**Table 1 ijms-23-11293-t001:** Quantitative image analyses of the effects of oligomycin and rotenone on mitochondrial membrane potential.

	6 h	20 h
Control	27.79 ± 0.64	27.19 ± 1.02
Oligomycin	14.87 ± 1.83 *	16.42 ± 1.04 *
Rotenone	18.33 ± 2.36	20.02 ± 2.56

* *p* < 0.05 compared with each control.

## Data Availability

The data are available from the corresponding author upon reasonable request.

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
