# Peer review of "Modulation of ATP Production Influences Inorganic Polyphosphate Levels in Non-Athletes’ Platelets at the Resting State"

_ijms, 2022, doi:10.3390/ijms231911293_

Round 1

Reviewer 1 Report

I read the manuscript "Modulation of ATP production influences inorganic polyphosphate levels in non-athletes' platelets at the resting state" with great interest. The Authors present results of oligomycin, rotenone and GlutaMax on the level of inorganic polyphosphate (polyP), ATP and NADH in resting platelets of non-athletes. The introduction, results and methods are presented in an accessible and comprehensible manner, but the discussion arouses much controversy. In my opinion it should be redrafted, especially in the part of "Energy metabolism and polyP turnover at cellular levels". In the line 205, authors postulate that several points needs to considered to progress the topics. It is difficult to interpret what the number (1) refers to. Then there are short inserts that remind more of brief notes or conclusions from the observation than a matter-of-fact discussion. There is no consistency here and it is difficult to follow the authors' lines of thought. The main focus should be on the findings in this study, compeered with findings in recent studies. It is not necessary to review which factors are known to affect ATP production. However, it is difficult to find a direct relationship between obtained results and discussed issues. Also, the novelty elements of this study are limited, especially since the authors primarily refer to their earlier observations in other publications.

I appreciate the Figure 10, where Authors propose mechanism of polyP conversion from ATP in resting platelets. According this figure rotenone and oligomycin should exert the same effects, but some differences were observed in results. Some discussion on this topic would be appreciated. Maybe use of others inhibitors of energy production (e.g. uncouplers (FCCP, DNP), cyanide, antimycin or use of the mitochondrial inhibitors in combination with deoxyglucose - inhibitor of glycolysis) could contribute to better understanding of the importance of inorganic polyphosphate in platelet.

Author Response

1) I read the manuscript "Modulation of ATP production influences inorganic polyphosphate levels in non-athletes' platelets at the resting state" with great interest. The Authors present results of oligomycin, rotenone and GlutaMax on the level of inorganic polyphosphate (polyP), ATP and NADH in resting platelets of non-athletes. The introduction, results and methods are presented in an accessible and comprehensible manner, but the discussion arouses much controversy. In my opinion, it should be redrafted, especially in the part of "Energy metabolism and polyP turnover at cellular levels". In the line 205, authors postulate that several points need to be considered to progress the topics. It is difficult to interpret what the number (1) refers to. Then there are short inserts that remind more of brief notes or conclusions from the observation than a matter-of-fact discussion. There is no consistency here and it is difficult to follow the authors' lines of thought. The main focus should be on the findings in this study, compeered with findings in recent studies. It is not necessary to review which factors are known to affect ATP production. However, it is difficult to find a direct relationship between obtained results and discussed issues. Also, the novelty elements of this study are limited, especially since the authors primarily refer to their earlier observations in other publications.

Response: Thank you for this comment. To help non-expert readers comprehend the purpose of this study, we devoted somewhat “excessive” space to describe the fundamentals and contexts of cellular energy metabolism and polyP turnover. However, because you deemed it “too much,” we have reduced the length of this subsection.

Regarding the numbers in the parentheses, we have listed the most important aspects of this study. “(1) Basic energy metabolism” is important to understand our study design. “(2) The chemical features of polyP microparticles” help understand the specificity of DAPI staining for fluorometric quantification and imaging. “(3) Chemical and biological stability of polyP and ATP” is closed related to our conclusive idea illustrated in Figure 10. Finally, “(4) Changes in NADH levels” is our response to the question “why is WST-8 reduction determined instead of ATP?” (based on feedback from the reviewers and readers of our previously published articles). Thus, we hope keep the numbers to improve the accessibility of our study to non-experts.

However, we also understand what you meant. Thus, we have reorganized this subsection and reduced the volume by deleting the section of NADH.

2) I appreciate the Figure 10, where Authors propose mechanism of polyP conversion from ATP in resting platelets. According to this figure, rotenone and oligomycin should exert the same effects, but some differences were observed in results. Some discussion on this topic would be appreciated. Maybe use of others inhibitors of energy production (e.g. uncouplers (FCCP, DNP), cyanide, antimycin or use of the mitochondrial inhibitors in combination with deoxyglucose - inhibitor of glycolysis) could contribute to better understanding of the importance of inorganic polyphosphate in platelet.

Response: Thank you for this comment. We have added a concise explanation of this illustration in the paragraph. We concur with you regarding the other inhibitors. We have compiled a list of inhibitors, including FCCP and the others you suggested, for constructing a working hypothesis. However, we chose the reagents for this study based on their commercial and legal availability. For example, it is challenging to use cyanide in our laboratory. Therefore, there is a need for further studies. However, contradictory data may be obtained due to the different sites of action. Therefore, to advance our understanding of platelet polyP metabolism, we think exploring additional inhibitors and describing their differences in detail is necessary.

Reviewer 2 Report

The manuscript by Ushiki et al., etitled Modulation of ATP production influences inorganic polyphosphate levels in non-athletes platelets at the resting state describes a series of pharmacological and imaging experiments examining the effect of electron transport chain and ATP synthase inhibitors on poly-P levels in human platelets. Based on these studies the authors propose that [ATP] and production may negatively regulate poly-P granule stores. Overall while these studies support previous studies by the authors there are several experimental controls re OxPhos inhibitors and overall platelet function that are required to support the overall conclusions.

Specific Comments:

1.       The authors claim that the PRP samples used in the experiments represent a resting population of platelets. The authors need to supply evidence of the basal platelet state ie., FACS analysis of Pac1 antibody binding, P-selectin expression, and/or imaging of platelet morphology (are the platelets predominantly discoid).

2.       What happens to poly-P levels as investigated using the DAPI staining protocol when platelets are directly activated using ADP or stronger agonsists such as thrombin?

3.       The authors demonstrate that in the presence of Rotenone neither ATP nor NADH levels changed significantly. The authors need to demonstrate the effect of Rotenone on electron transport chain function in platelets. Respirometry data of the platelet suspension would confirm the level of inhibition.

4.       What is the respiratory coupling ratio in the PRP samples utilised? How does this impact poly-P levels.

5.       Platelet function in isolated PRP substantially degrades after 4 hours post-isolation. Functionally, degradation is in part due to low level ADP secretion leading to sub-maximal activation. What impact do ADP/ATP scavengers have on overall pol-P levels +/- electron transport chain inhibition? How can the authors rule out an impact of sub-maximal ADP driven platelet activation in 20h samples?

Author Response

The manuscript by Ushiki et al., entitled Modulation of ATP production influences inorganic polyphosphate levels in non-athletes platelets at the resting state describes a series of pharmacological and imaging experiments examining the effect of electron transport chain and ATP synthase inhibitors on poly-P levels in human platelets. Based on these studies the authors propose that [ATP] and production may negatively regulate poly-P granule stores. Overall while these studies support previous studies by the authors there are several experimental controls re OxPhos inhibitors and overall platelet function that are required to support the overall conclusions.

Response: Thank you very much for your insightful comments and suggestions. We concur with you; however, we do not have any device for examining platelet respiration. Moreover, we have been given only ten days for the revision (at least 4 out of 10 days are needed for English editing). Thus, we must revise our manuscript using the available information.

Specific Comments:

1) The authors claim that the PRP samples used in the experiments represent a resting population of platelets. The authors need to supply evidence of the basal platelet state ie., FACS analysis of Pac1 antibody binding, P-selectin expression, and/or imaging of platelet morphology (are the platelets predominantly discoid).

Response: Few studies have examined platelet activation following blood collection; however, platelets are likely to activate and converge rapidly after collection. In addition, platelets are known to be activated in patients with cardiovascular disorders, but only healthy subjects were examined in this study. Therefore, we presumed that platelets entered a resting phase in the presence of a Ca2+-chelator. Please see the article below (Abrams et al., Blood 1990).

https://www.sciencedirect.com/science/article/pii/S0006497120858285?via%3Dihub

Furthermore, we did not assess activated platelet levels in this investigation. However, we mentioned that most of this population consists of platelets at rest, based on our previous findings referred below.

Please find these articles.

  • Aizawa et al. Biomedicines 2020, 8(3), 42 https://www.mdpi.com/2227-9059/8/3/42/htm
  • Sato et al. Int. J. Mol. Sci. 2021, 22(3), 1040 https://www.mdpi.com/1422-0067/22/3/1040

Please see also:

  • Tsujino et al., Dent. J. 2019, 7(4), 109 https://www.mdpi.com/2304-6767/7/4/109
  • Nakamura et al., International Journal of Implant Dentistry volume 6, Article number: 74 (2020) https://journalimplantdent.springeropen.com/articles/10.1186/s40729-020-00270-2

2) What happens to poly-P levels as investigated using the DAPI staining protocol when platelets are directly activated using ADP or stronger agonists such as thrombin?

Response: Thank you for your question. In a recent study [Sato et al., IJMS 2021, 22(3), 1040], we examined the effects of CaCl2 on platelet polyP levels by visualization and observed platelet aggregation and concomitant reduction in platelet polyP. In addition, we simultaneously examined the effects of ADP and thrombin. This reduction was less than that for CaCl2, but the agonists reproducibly reduced platelet polyP. However, this finding is not closely related to the purpose of this study, and thus we decided to exclude this result.

3) The authors demonstrate that in the presence of Rotenone neither ATP nor NADH levels changed significantly. The authors need to demonstrate the effect of Rotenone on electron transport chain function in platelets. Respirometry data of the platelet suspension would confirm the level of inhibition.

Response: Thank you for your valuable suggestion. Currently, we cannot use a respirometer or similar device on campus. However, if it becomes feasible, we aim to verify it.

4) What is the respiratory coupling ratio in the PRP samples utilised? How does this impact poly-P levels.

Response: As described above, we have no means to determine the respiratory coupling ratio. However, this would be a highly effective tool for elucidating the mechanism of the ATP–polyP conversion system.

5) Platelet function in isolated PRP substantially degrades after 4 hours post-isolation. Functionally, degradation is in part due to low level ADP secretion leading to sub-maximal activation. What impact do ADP/ATP scavengers have on overall pol-P levels +/- electron transport chain inhibition? How can the authors rule out an impact of sub-maximal ADP driven platelet activation in 20h samples?

Response: We appreciate the reviewer pointing out this critical point. Because centrifugation activates platelets, we cannot rule out the possible leakage of platelet ADP (and polyP and other growth factors) after preparation and incubation. Thus, to reduce this possibility, we used Ca- and Mg-free PBS instead of the standard Tyrode solution and room temperature instead of 37 °C.

Regarding the ATP scavenger, cyanide can be used. However, this reagent is strictly restricted in our lab due to legal constraints. To obtain permission, we need much more time and should provide more documentation. Thus, designing experiments using the limited set of available tools is essential.

Using SEM, we could compare the morphology of freshly isolated platelets with that of incubated platelets. Both populations were heterogeneous and consisted of disk-like platelets (resting; majority), platelets with numerous protrusions (activated), aggregated platelets (activated), spherical platelets, and readily adherent platelets. We did not observe any substantial morphological differences between these populations. When a promising quantification approach is developed in the future, we hope to reevaluate our results to address your question.

Round 2

Reviewer 2 Report

My question re "The authors demonstrate that in the presence of Rotenone neither ATP nor NADH levels changed significantly. The authors need to demonstrate the effect of Rotenone on electron transport chain function in platelets. Respirometry data of the platelet suspension would confirm the level of inhibition."

Has not been naswered experimentally.

I believe the authors need to be given sufficient time to conduct the relevant control experiments and include this data before I can deem it acceptable for publication.

Author Response

Dear Guest editors,

Thank you very much for giving us a second opportunity for major revision. However, to the best of our knowledge, there are no micro-respirometry systems at our university or in neighboring institutions. In addition, we could not find appropriate outsourcing agents that could offer affordable prices. Therefore, we chose an indirect method to demonstrate inhibition of the platelet respiratory chain.

  • Reviewer 2

1. My question re "The authors demonstrate that in the presence of Rotenone neither ATP nor NADH levels changed significantly. The authors need to demonstrate the effect of Rotenone on electron transport chain function in platelets. Respirometry data of the platelet suspension would confirm the level of inhibition." Has not been naswered experimentally.

Response: Thank you for this comment. We understand that such supporting data are important for reinforcing this conclusion. However, as described in the response to your comments in the first round, we could not find any microrespirometry systems suitable for platelets in our local area. In addition, such a rare device is not commercially available, and outsourcing agents that provide services using this device are not found in our country. Thus, we adopted an indirect alternative method to examine platelet respiration. These data are the best we can offer at present.

2. I believe the authors need to be given sufficient time to conduct the relevant control experiments and include this data before I can deem it acceptable for publication.

Response: We inquired about the special issue and whether it would be open for submission post two months (sufficient time for us to revise any data). However, we were informed that according to the publisher’s policy, a two-month period is too long to handle a revised version. As a result, only a month was provided for this revision.

With regard to English editing, we have always secured professional English editing service from Editage, prior to every submission. However, we did not evaluate the sophistication of the revised manuscript. For us, as non-native speakers, language is often a major barrier that we have made our best efforts to overcome.

Round 3

Reviewer 2 Report

Overall the authors have attempted to address my issues re the efficacy of respiratory chain inhibitors. However, I think for clarity the authors should show fluorescence quantitation for images associated with Figures 2 & 4 across all 4 replicate experiments showing mean +/-SD.

Author Response

Reviewer 2

Overall the authors have attempted to address my issues re the efficacy of respiratory chain inhibitors. However, I think for clarity the authors should show fluorescence quantitation for images associated with Figures 2 & 4 across all 4 replicate experiments showing mean +/-SD.

Response: Thank you for the valuable advice. We performed image analysis of 4 images in each group for quantitative comparisons. The non-parametric analysis demonstrated significant differences between the control and the oligomycin-treated group in both 6h and 20h. Rotenone reproducibly reduced the MT-1-specific luminance, but significant differences were not observed in either 6h or 20h. Please see Table 1.